# Reconnaissance of Surface Water Estrogenicity and the Prevalence of Intersex in Smallmouth Bass (*Micropterus Dolomieu*) Inhabiting New Jersey

**DOI:** 10.3390/ijerph17062024

**Published:** 2020-03-19

**Authors:** Luke R. Iwanowicz, Kelly L. Smalling, Vicki S. Blazer, Ryan P. Braham, Lakyn R. Sanders, Anna Boetsma, Nicholas A. Procopio, Sandra Goodrow, Gary A. Buchanan, Daniel R. Millemann, Bruce Ruppel, John Vile, Brian Henning, John Abatemarco

**Affiliations:** 1US Geological Survey, Leetown Science Center, Kearneysville, WV 25430, USA; vblazer@usgs.gov (V.S.B.); ryan_braham@fws.gov (R.P.B.); lsanders92@gmail.com (L.R.S.); 2US Geological Survey, New Jersey Water Science Center, Lawrence, NJ 08648, USA; ksmall@usgs.gov (K.L.S.); aboetsma@usgs.gov (A.B.); 3New Jersey Department of Environmental Protection, Division of Science and Research, Trenton, NJ 08625, USA; Nick.Procopio@dep.nj.gov (N.A.P.); sandra.goodrow@dep.nj.gov (S.G.); Gary.Buchanan@dep.nj.gov (G.A.B.); Dan.Millemann@dep.nj.gov (D.R.M.); Bruce.Ruppel@dep.nj.gov (B.R.); 4New Jersey Department of Environmental Protection, Division of Water Monitoring and Standards, Trenton, NJ 08625, USA; john.vile@dep.nj.gov (J.V.); brian.henning@dep.nj.gov (B.H.); john.abatemarco@dep.nj.gov (J.A.)

**Keywords:** intersex, endocrine disruption, estrogen, fish, smallmouth bass, bioassay, land use

## Abstract

The observation of testicular oocytes in male fishes has been utilized as a biomarker of estrogenic endocrine disruption. A reconnaissance project led in the Northeastern United States (US) during the period of 2008–2010 identified a high prevalence of intersex smallmouth bass on or near US Fish & Wildlife Service National Wildlife Refuges that included the observation of 100% prevalence in smallmouth bass males collected from the Wallkill River, NJ, USA. To better assess the prevalence of intersex smallmouth bass across the state of New Jersey, a tiered reconnaissance approach was initiated during the fall of 2016. Surface water samples were collected from 101 (85 river, 16 lake/reservoir) sites across the state at base-flow conditions for estrogenicity bioassay screening. Detectable estrogenicity was observed at 90% of the sites and 64% were above the US Environmental Protection Agency trigger level of 1 ng/L. Median surface water estrogenicity was 1.8 ng/L and a maximum of 6.9 ng/L E2Eq_BLYES_ was observed. Adult smallmouth bass were collected from nine sites, pre-spawn during the spring of 2017. Intersex was identified in fish at all sites, and the composite intersex prevalence was 93.8%. Prevalence across sites ranged from 70.6% to 100%. In addition to intersex, there was detectable plasma vitellogenin in males at all sites. Total estrogenicity in surface water was determined at these fish collection sites, and notable change over time was observed. Correlation analysis indicated significant positive correlations between land use (altered land; urban + agriculture) and surface water estrogenicity. There were no clear associations between land use and organismal metrics of estrogenic endocrine disruption (intersex or vitellogenin). This work establishes a baseline prevalence of intersex in male smallmouth bass in the state of New Jersey at a limited number of locations and identifies a number of waterbodies with estrogenic activity above an effects-based threshold.

## 1. Introduction

Endocrine disruption is a specific form of toxicity whereby anthropogenic or natural compounds, defined as endocrine disruptors (EDs), potentiate unfavorable health effects by disrupting normal organismal hormone signaling networks [1]. Endocrine-disrupting chemicals have been shown to affect endocrine pathways in fish and wildlife inhabiting natural environments [2,3,4]. Moreover, EDs and endocrine activity in surface waters is widespread and even observed in remote aquatic environments [5,6,7,8,9,10]. Although EDs have been documented extensively in natural systems, establishing cause-and-effect relationships with biological observations is typically not possible and the fulfillment of Hill’s Criteria or simple weight-of-evidence associations are the most satisfactory observational support [2,11,12]. Given the global ecological health implications of ED exposure, considerable efforts have been endeavored to develop strategies that evaluate and predict the significance of exposures [13,14,15].

Fish are routinely used as sentinels of endocrine disruption in the aquatic environment. Evidence of estrogenic endocrine disruption has been reported in fishes since the 1990s [16,17]. Biological evidence of estrogenic endocrine-disrupting chemical (EEDC) exposure in gonochoristic male fishes generally includes the morphological presentation of testicular oocytes (TOs), also known as intersex. Numerous, comprehensive surveys have been conducted that evaluate the prevalence of TOs in smallmouth bass (*Micropterus dolomieu*) and largemouth bass (*Micropterus salmoides*) [18,19,20,21,22,23,24,25,26,27,28]. Many of these field evaluations have identified significant associations between the incidence and severity of TOs with land use or contaminants [19,25,29]. Yet others have noted that such associations are sometimes less clear [26,28]. 

During the period 2008–2010, a reconnaissance of US Fish & Wildlife Service Northeast Region National Wildlife Refuges was conducted that used the prevalence of TOs in male bass as evidence of EEDC exposure. Results from that research identified male smallmouth bass with TO collected from New Jersey waters [24]. More specifically, a prevalence of 100% intersex was observed in smallmouth bass samples from the Wallkill River, NJ, USA collected during the fall. The sample size (*n* = 4) of that survey was small and to ascertain a more comprehensive prevalence of this condition across the state of NJ, USA, we endeavored a strategic sampling approach. 

The Ambient Surface Water Quality Monitoring Network is a cooperative effort between the Bureau of Freshwater and Biological Monitoring in the New Jersey Department of Environmental Protection (NJDEP) and the U.S. Geological Survey, New Jersey Water Science Center (NJWSC). The network was established in 1976 to determine the status and trends of water quality in the surface waters of New Jersey. The data collected were intended to support the departmental responsibilities under the federal Clean Water Act of 1972, the federal Safe Drinking Water Act of 1974, and the National Shellfish Sanitation Program, in addition to other programs. The Division of Science and Research, within the NJDEP, provides an extension of technical expertise in identifying emerging issues and forming experimental designs to ascertain the impact on the waters of New Jersey. Here, we leveraged well-established water sampling sites across the state of New Jersey with a yeast-based screening assay for estrogenicity and curated fish assemblage databases to identify locations for biological sampling across a gradient of predicted estrogen exposure [5,9,10,24]. Subsequently, smallmouth bass were collected and biomarkers of EEDC exposure were assessed. 

## 2. Methods

### 2.1. Site Selection

#### 2.1.1. Sampling Locations

We targeted 101 sites (rivers and lakes/reservoirs) during the fall of 2016 for the preliminary estrogenicity assessment. Sites were selected utilizing compiled results of previous monitoring studies (chemical and biological), current sampling networks [30], and other areas of concern (Figure 1). Based on results from discrete grab surface water samples (methods below), in combination with land-use gradients, spatial distributions and the availability of viable fish populations, we identified 9 sites for the collection of smallmouth bass. These locations included reservoirs, lakes and sites along the Delaware River. For more information on sampling locations, see [31].

#### 2.1.2. Water Sampling and Estrogenicity Bioassay

During the fall of 2016, discrete grab water samples were collected and analyzed for total estrogenicity using a bioluminescent yeast screen. Samples were collected under base-flow conditions and these data were, in part, utilized to identify sites for the collection of smallmouth bass to further investigate evidence of adverse biological effects. Duplicate samples were collected for 19% of the collection sites. Water samples were collected again at the time of fish sampling during April and May of 2017 to compliment biological datasets, and then again later in the fall of 2017 to establish a repeated seasonal measure of estrogenicity. Surface water was collected in pre-cleaned, 1 L amber glass bottles, handled and extracted as previously described [32]. Total estrogenicity was determined for extracted water samples using strain BLYES [33], as described previously [32]. The reporting limit for discrete water samples was 0.23 ng/L E2Eq_BLYES_. 

#### 2.1.3. Fish Collection and Processing

During the spring of 2017, smallmouth bass were sampled from 9 locations between April and May (Figure 1). Fish were collected from each of the 9 locations within as maximally compressed time window as logistically possible to circumvent natural biological differences based on seasonal change (Table 1). We attempted to capture 10 female and 10 male smallmouth bass (>250 mm) using pulsed-DC electroshocking at each sample location. All fish were collected pre-spawn. Fish collection, handling and euthanasia protocols were approved by the U.S. Geological Survey, Leetown Science Center Institutional Animal Care and Use Committee. Fish were maintained in aerated coolers containing water from the collection site prior to processing. Fish necropsy and processing was conducted, as described previously [34]. Briefly, euthanasia was achieved using a lethal dose (300–500 mg/L) of tricaine methanesulfonate (Finquel, Argent Laboratories, Redmond, WA). Fish were bled from the caudal vasculature, and blood was expressed into microcentrifuge tubes pre-loaded with 1000 units of sodium heparin. Tubes were inverted to mix and placed on wet ice. Blood was centrifuged within four hours of collection using a portable myFUGE^TM^Mini (Benchmark Scientific, Sayreville, NJ, USA) for 10 min at 2000× *g* for plasma separation. Plasma was removed, aliquoted into cryovials and stored at −80 °C. Each fish was weighed (to the nearest g), measured (to the nearest mm), observed for gross lesions and abnormalities, and the liver and gonad were removed and weighed to the nearest 0.01 g. A portion of the liver was preserved in RNA*later*^®^ for molecular analyses. Lapillus otoliths were extracted and prepared to determine fish age. Condition factor (Ktl) was determined using the formula: ((body weight − gonad weight in g) / length^3^ in mm) × 10^5^. The gonadosomatic index was calculated using the formula: (gonad weight in g/ body weight in g) × 100. Gonads were removed and fixed in Z-Fix™ (Anatech Ltd., Battle Creek, MI, USA). Routine processing using paraffin-based methods and hematoxylin and eosin staining was used to prepare tissues for histopathological evaluation [35]. 

#### 2.1.4. Reproductive Endpoints

Intersex severity was assigned using methods described elsewhere [19]. Plasma vitellogenin (Vtg) concentrations were determined using a direct enzyme-linked immunosorbent assay (ELISA) with monoclonal antibody 3G2 (Caymen Chemical, Ann Arbor, MI, USA) as previously described [24,29,36].

#### 2.1.5. Transcript Abundance Analysis

Total RNA was extracted from liver tissues stabilized in RNA*later.* Extraction, cDNA synthesis and reverse transcription quantitative PCR (RT-qPCR) for vitellogenin Aa (*vtgAa*) transcripts were carried out, as described previously [37]. Elongation factor alpha (*EF1α*) was used as a housekeeping gene to normalize transcript abundance. All RT-qPCR was conducted on a Vii^TM^ 7 Real-Time PCR System (Applied Biosystems, Willmington, DE, USA).

#### 2.1.6. Land Use Summary

A drainage basin was delineated for each sampling site using ArcGIS. The 2011 National Land Cover Database (NLCD 2011) from the Multi-Resolution Land Characteristics Consortium was used to determine the statistical breakdown of land use types within each site-basin. Data regarding the geospatial distribution of discharges to surface water and ground water in each basin were also compiled. 

Discharges to surface water and ground water require a New Jersey Pollutant Discharge Elimination System (NJPDES) permit through the New Jersey Department of Environmental Protection’s Division of Water Quality. Downloadable shapefiles for both surface water and ground water NJPDES permit sites are available on the NJDEP Bureau of GIS webpage [38]. Using these data, the number and type of surface water and ground water discharges in each basin were summarized using ArcGIS. 

The number of contaminated sites, as defined by NJDEP, and the number of landfills were summarized for each basin using GIS. These data are also available on the NJDEP Bureau of GIS webpage. 

### 2.2. Statistical Analyses 

Data were tested for normality (Shapiro–Wilk test) and homogeneity of variance (Levene’s test). Differences in estrogenic activity between water collected from rivers versus impoundments were evaluated with an unpaired t-test and Welch’s correction. Differences in detection frequency between these water sources were compared via Fisher’s Exact Test. Differences in composite estrogenicity across years for the subset of nine fish sampling sites were compared using the Friedman test and Dunn’s multiple comparison. 

Differences in intersex prevalence across sites were evaluated using Fisher’s Exact Test. Intersex severity was compared across sites using one-way ANOVA followed by Tukey’s post-hoc analysis. Plasma Vtg and hepatic Vtg transcript abundance were compared across sites separately for males and females using the Kruskal–Wallis test followed by Dunn’s multiple comparison post-hoc analysis. Plasma Vtg was compared between males and females at each site using the Mann–Whitney U test. Associations between age, intersex severity, plasma Vtg and *vtgAa* were analyzed using Spearman rank correlation analysis. 

Relations among estrogenic activity, intersex severity, plasma Vtg and liver *vtgAa* and a suite of watershed-level land-use predictor variables were evaluated using Spearman rank-order correlations. The watershed-level predictor variables included major land-use categories (as a percent), watershed area, the number of groundwater and surface water discharge permits, the number of contaminated sites and the number of landfills (Table 2 and Table 3). For all statistical analyses, any estrogenicity value reported as below the limit of quantitation was assigned a value of 0.1 ng/L E2Eq_BLYES_.

## 3. Results

### 3.1. Surface Water Estrogenicity

Discrete grab water samples were collected at 101 sites (85 river, 16 lake/reservoir; herein referred to as impoundments) during the fall of 2016. Estimated estrogenicity ranged from below detection to a maximum of 6.9 ng/L of E2Eq_BLYES_. The median estrogenicity was 1.5 ng/L. Estrogenic activity above the limit of quantitation was observed in 94% of the samples across all sites. Estrogenic activity was at or above the US Environmental Protection Agency (USEPA) effects-based trigger (EBT) value of 1 ng/L of E2Eq_BLYES_ at 64% of the sites sampled. There were no statistical differences between estrogenic activity between river or impoundments during fall 2016, nor were there differences in detection frequency.

Nine sites were selected for fish sampling and estrogenic activity was determined at these locations in the spring of 2017 and fall of 2017 in addition to the previous fall 2016 sampling. Statistical differences between site per season or across season at specific sites could not be assessed due to the sampling design, but qualitatively there appeared to be differences (Figure 2A). Composite samples based on season of collection identified seasonal differences in estrogenic activity. Notably, estrogenic activity was statistically higher in the fall of 2016 compared to the fall of 2017 (*p* < 0.001). Estrogenicity and site metadata are available at https://doi.org/10.5066/P9LZKY6Z [30].

### 3.2. Prevalence and Severity of Testicular Oocytes 

A total of 174 (M = 115; F = 59) fish were collected for analysis (Table 1). Intersex (testicular oocytes) was identified in male smallmouth bass at all sites during all sample collections (Figure 2B). The composite intersex prevalence across sites was 93.8%. On a per site basis, intersex prevalence ranged from 70.6% to 100%, and the greatest intersex severity (ISS) for an individual fish at a particular site (Yards Creek Reservoir) was 2.4. Intersex severity was statistically lower in fish collected from Boonton Reservoir compared to Splitrock Reservoir (*p* < 0.001), Canister Reservoir (*p* < 0.001) and Yards Creek Reservoir (*p* = 0.002). There were no statistical differences in intersex severity across other sites. Notably, there were also no significant differences in the prevalence of intersex across sites. Intersex severity was not correlated with age, length, weight or condition factor.

### 3.3. Plasma Vitellogenin and Differential Expression of Hepatic Transcripts

Plasma vitellogenin was detected in all fish (both male and female) at all collection sites. Plasma vitellogenin was statistically greater in females than males inhabiting the same location (*p* = 0.005). Differences in plasma Vtg were observed in males and females across sites (Figure 3). Plasma Vtg was lower in females collected from Splitrock Reservoir compared to those from Yards Creek Reservoir (*p* = 0.003). There were no differences in female plasma Vtg across other sites. Differences in transcription of the *vtgAa* gene were observed in females across sites (*p* < 0.0001). Composite analysis of the relationship between plasma vitellogenin and hepatic transcription of *vtgAa* identified a significant positive relationship (*n* = 59; ρ = 0.301; *p* = 0.021). 

Plasma vitellogenin was detected in all males and ranged from 119 to 918 µg/mL across all sites. Median plasma Vtg from these sites ranged from 272–703 µg/mL observed at Boonton and Round Valley Reservoir, respectively. Statistical differences were observed in plasma Vtg across sites (*p* < 0.001; Figure 3). Correlation analysis identified significant positive correlations between plasma Vtg and age (*n* = 111; ρ = 0.421; *p* < 0.001) and intersex severity (*n* = 112; ρ = 0.261, *p* = 0.006). There was a significant negative correlation between plasma Vtg and condition factor (*n* = 115; ρ = −0.330; *p* < 0.001).

Similarly, there were statistical differences observed in hepatic expression of *vtgAa* in male fish. Unlike the relationship between plasma Vtg and hepatic transcription observed in females, there was no significant relationship between these measures in male fish. Significant positive correlations were identified between hepatic *vtgAa* expression and weight (*n* = 114; ρ = 0.188; *p* =0.045), length (*n* = 114; ρ = 0.204; *p* = 0.029), gonad weight (n = 114; ρ = 0.243; *p* = 0.009) and GSI (*n* = 114; ρ = 0.306; *p* < 0.001).

### 3.4. Associations with Land Use

Spearman rank-order correlation analyses were utilized to explore associations between biological endpoints in males and land use. Given that there were no significant differences in intersex prevalence across sites, we did not explore such associations with intersex metrics to avoid spurious associations. Associations were identified between land use and male plasma Vtg or liver transcript expression of *vtgAa* (Table 3). Percent cultivated crops was the only land-use predictor that was not statistically correlated with male plasma vitellogenin. Interestingly, the other land uses all negativity correlated with plasma Vtg. Expression of *vtgAa* was positivity correlated with percent urban and percent altered land use (urban + agriculture) and negatively correlated with percent forest. 

Surface water estrogenicity from the fall of 2016 sampling was statistically correlated with land use (Table 2). When all water samples were collectively analyzed (rivers and impoundments), percent urban (*n* = 101; ρ = 0.206; *p* = 0.039) and percent altered land (*n* = 101; ρ = 0.306; *p* = 0.002) were positively correlated with estrogenicity. Notably, when rivers and impoundments were investigated separately, it was clear that the observed associations were driven by the river samples. No statistically significant relationships were observed between land use and estrogenicity within impoundments. Positive correlations were identified among river sites between estrogenicity and percent urban (*n* = 85; ρ = 0.277; *p* = 0.011), percent altered (*n* = 85; ρ = 0.402; *p* = 0.002), number of contaminated sites (*n* = 85; ρ = 0.253; *p* = 0.020), and number of surface water discharge permits (*n* = 85; ρ = 0.249; *p* = 0.023). 

## 4. Discussion

The primary objective of this survey was to determine the prevalence and severity of estrogenic endocrine disruption in fish within the state of New Jersey using smallmouth bass as an indicator in relation to landscape drivers and surface water estrogenicity. Sites were selected based on the gradient of estrogenicity values determined during base-flow conditions in the fall prior to biological sampling using an in vitro estrogenicity bioassay. The rationale for using this selection process was that it would allow capturing the range of intersex within smallmouth bass populations based on the assumption that surface water estrogenicity was predictive of intersex prevalence and severity. The application of in vitro bioassay screening has led to an unprecedented attribution of biological potential ascribed to surface water samples during recent years [5,9,39,40,41,42]. In the absence or in conjunction with analytical chemistry, such datasets have provided biological-proxy data without the need to sacrifice resident biota and have the potential to inform predictive modelling to guide biological sampling site selection [43]. Such evaluations have yielded data that facilitate analyses over greater spatial scales than is possible using a resident proxy organism. The goal of surface water sampling was to inform smallmouth bass site selection utilizing bioassay estrogen equivalents data and assess relations with specific landscape drivers across a range of sites and biological activity, but not to measure specific EEDCs. 

Detectable estrogenicity was observed at 90% of the sites and 64% were above the USEPA EBT of 1 ng/L [5]. Median surface water estrogenicity was 1.8 ng/L and a maximum of 6.9 ng/L E2Eq_BLYES_ was observed. Recently an EBT range of 0.1–0.5 ng/L E2Eq has been proposed [44]. Based on this EBT range, all the samples in the current dataset with quantifiable estrogenicity (90%) are within this trigger-level range. Previous studies have identified highly variable estimates of estrogenic activity in stream water. Estrogenicity of surface water has been sampled from six countries; Germany, Australia, France, South Africa, The Netherlands, and Spain ranged from <0.1–0.31 ng/L E2Eq [43], but have been reported as high as 10 ng/L in the upper Chesapeake Bay watershed (USA) and 13.8 ng/L in Xiamen (China) [32,39,45]. Estrogenicity of other surface waters with documented intersex smallmouth bass is often considerably lower than values measured in these New Jersey waters [24,29]. However, there is no species-specific diagnostic estrogen equivalent value ascribed to intersex induction.

This study identifies differences in estrogenicity across season for the repeated measures at the nine biological sampling sites. The salient fact here being that while estrogen equivalent values during a given sampling event may be relevant for that specific date in time, it may not adequately represent the dynamic range or mean at a given location. This range is likely to be variable and differ across locations based on sources and both mechanisms and dynamics of transport and fate. This is not unexpected in lotic riverine systems, given continuous flow and flushing, but variable measures were noted in water collected from impoundments as well. Considerably less information is available on estrogen equivalents in more stagnant water bodies like lakes, impoundments or ponds but the few studies that exist note detectable levels of both estrogenic activity and EEDCs [46,47]. The biological endpoints measured in the smallmouth bass are essentially temporal clocks. Expression of *vtgAa* is utilized as measure of exposure to estrogens within multiple hours or days prior to capture; plasma Vtg is a measure of estrogen exposure many days to weeks prior to capture; and the observation of testicular oocytes captures a larger temporal window. If, indeed, the high intersex prevalence noted in the smallmouth bass evaluated here is a result of EEDC exposure, it is possible that the bioassay data poorly predicted this observation due to an artifact of the temporal snapshot. Notably, surface water estrogenicity in this survey was not predictive of intersex prevalence or severity. Based on the temporal variation observed in surface water estrogenicity and the limited number of sites sampled for intersex screening this is not surprising. That stated, the presence of endocrine active chemicals in sediment and not surface water has been shown to be more predictive of intersex in *Micropterus* spp. [25]. Perhaps evaluating estrogenicity of sediment samples could have provided predictive values. However, given that there was essentially no difference in intersex prevalence or severity across the limited number of sites sampled (the exception being Boonton Reservoir), predictive relationships cannot be expected. 

Intersex prevalence was high across all sites and it is unclear what specific factors may have influenced this. It should be emphasized that most of the sample locations in this study were impoundments such as reservoirs and lakes. Smallmouth bass are non-native to waters outside the upper-Mississippi River, Ohio River and tributaries of the Lower Missouri River. While these fish have naturalized in many Eastern river systems in the United States, there are active stocking programs in New Jersey which, in part, support smallmouth bass populations in some of these impoundments including Yards Creek Reservoir, Canistear Reservoir, Splitrock Reservoir and Echo Lake. However, only Yards Creek and Splitrock Reservoirs have been stocked since 2004 (i.e., in 2011 and 2012, respectively) [48]. It is possible that exposure events such as hatchery feed with high estrogens, occurrence of periodic algal blooms and/or increases in phytoplankton mass thought to contain natural estrogens [49,50] may explain the consistent observation of high intersex. However, it is unlikely that hatchery feed was a cause as few if any fish sampled during this reconnaissance were hatchery reared. In addition, there is a lack of intersex data for smallmouth bass that inhabit impoundments. As a result, it is unclear whether comparing intersex prevalence and severity in smallmouth bass inhabiting impoundments to those from lotic systems is appropriate for contrast. For instance, high intersex prevalence (70%) was observed in smallmouth bass inhabiting Lake Umbagog located at the border of New Hampshire and Maine (USA) and is relatively isolated from most anthropogenic land use [24]. Fate and transport of estrogen active compounds in these systems is also not well studied and is likely dissimilar to that in lotic systems. Urban land use, the number of surface water discharge permits (including wastewater discharges) and the number of known contaminated sites were all positively correlated to estrogenicity in our study. Although we did not have enough sites to statistically compare intersex data to known point sources, similar to other studies [51,52], the presence of these point sources could have contributed to intersex observed across our study area. However, for several of the impoundments (Echo Lake, Canistear Reservoir and Splitrock Reservoir), there are no well-defined point sources that would contribute endocrine active chemicals into the impoundments where fish were sampled during the New Jersey reconnaissance study. It is possible that point sources contributed EEDCs to some of the impoundments, but the presence or absence of a defined point source clearly does not fully explain the prevalence of intersex in all impoundments. Of note, several studies have identified the presence of natural estrogens (flavonoids) from plants, specifically plankton and algal blooms, in both freshwater and marine environments [47,49], with notable induction of vitellogenin in laboratory reared fish [50]. This class of compounds may warrant further investigation.

Vitellogenin expression (plasma protein and hepatic mRNA) is perhaps the most widely adopted endpoint indicating estrogen exposure in male fishes [4,53]. We observed statistical differences in both plasma vitellogenin and liver mRNA expression across some sites in the current study. In general, plasma vitellogenin was high at all sites relative to previously reported plasma vitellogenin concentrations in smallmouth bass [24,28,29,54]. Studies have noted that exposure to nitrate leads to increased vitellogenin [55,56]. While nitrate has been recognized as a putative endocrine disruptor for years, it is still understudied as a driver or risk factor [57,58]. Although, many of our riverine sampling locations have a long history of nitrate measurements, limited data are available at our impoundment sites and we did not measure nitrate during our sampling making it impossible to assess relationships among vitellogenin and nitrate. Spawning fish have also been identified as a source of estrogens in the aquatic environment [59]. Smallmouth bass are piscivorous and it seems plausible that ingestion of forage during high estrogen stages of the reproductive cycle may explain increases in baseline vitellogenin. Interestingly this is not a commonly acknowledged or investigated exposure route, even though ingestion of a single female prey item may lead to an exposure of many micrograms or estrogens.

Correlative associations were not observed between surface water estrogen equivalents on the day of sample collection and either male plasma vitellogenin or liver mRNA. When these metrics were analyzed for associations with land use, some highly contrasting and difficult to explain observations were made. Male plasma vitellogenin was negatively correlated with land use evaluated except percent cultivated crops, for which there was no association. Associations between hepatic *vtgAa* expression was positively correlated with percent altered land and negatively associated with percent forest. The latter association seems more intuitive, but, in general, it is difficult to make strong inferences from these data at a limited number of sites. Notably, plasma vitellogenin was positively associated with intersex severity, and negatively associated with condition factor. This may be an informative association, but the temporal window here is limited. A weak positive correlation between intersex severity and plasma vitellogenin has been reported previously in this species [60].

The composite prevalence of testicular oocytes across all sites for this study was 94%. Qualitatively, this is much higher than the prevalence reported in the National Wildlife Refuge reconnaissance study (composite 65% between sites). At seven of the nine sites here, intersex was observed in 100% of the fish. At the remaining two sites, intersex was detected in 70.6% and 91.7% of fish, respectively. Seasonal differences in intersex prevalence have been reported, and a smallmouth bass intersex study in Vermont reported that testicular oocyte prevalence and severity was lower in the spring than fall and can change over time independent of year-class [27,28]. Thus, the prevalence observed here may be conservative. Intersex prevalence greater than 90% is not unprecedented; however, it is considerably greater than the suggested baseline prevalence of 10%–14% based on values derived from the Ohio River drainage [19,24,28,54]. Of note, the intersex severity observed in smallmouth bass from the Ohio River drainage in the period of 2007–2010 was approximately 0.16, which again is considerably lower than the composite average intersex severity of 1.04 observed here [54]. Intersex severity has been identified as a predictor of decreased sperm quality in smallmouth bass, but a severity level has not been experimentally derived [29]. At present intersex severity values are not diagnostic of a specific biological outcome. Comprehensive investigations of roach (*Rutilus rutilus*) suggest that the prevalence of intersex may be 0.50% under natural conditions [61]. This is lower than the baseline frequency of intersex in smallmouth bass, but it is the most relevant species for comparison. Of note, the severity of intersex in the roach is associated with impaired sperm function. In the modern day, there are few aquatic systems inhabited by smallmouth bass that are genuinely pristine or unimpacted. Observations of intersex prevalence in smallmouth bass seem to parallel that of roach, which are commonly used as a sentinel of EEDC–associated disruption in European aquatic habitats, and testicular oocytes are associated with reduced fertility [62,63]. It has been noted that intersex severity increases with age in some species [64] but we did not observe this association in this study.

While intersex has been monitored in black bass for over a decade, the specific drivers that lead to the induction of testicular oocytes in the smallmouth bass have yet to be resolved. This likely indicates that there is no one simple driver and that expression of this condition is the culmination of numerous factors that likely encompass biotic and abiotic risk factors of natural and anthropogenic origin. While land use and organic-chemical signatures have been identified as predictive variables of intersex in smallmouth bass, there appear to be differences across geographical river basins that suggest there is no one-size-fits-all model [25,26,27,29]. In summary of the observation from this current reconnaissance in New Jersey waters, there is clearly a high prevalence of testicular oocytes in smallmouth bass populations. Surface water estrogenicity is at or above the EBT at a frequency of 90% during base-flow conditions and there is great variation in surface water estrogenicity over time. If indeed estrogenicity or other chemical stimuli are associated with intersex, it is possible that the timing of our collections missed critical windows of exposure. Future work that includes the analysis of estrogenicity at more frequent intervals and analytical chemistry may lead to more definitive insights. Notably, there is evidence that intersex in the environment is not permanent. Comprehensive research in the Grand River (Ontario, Canada) with the rainbow darter (*Etheostoma caeruleum*) identified that removal of feminizing drivers (WWTP upgrades in this case) led to a decline in intersex over time [50]. There is also evidence that intersex prevalence and severity can decrease rapidly (over just two reproductive cycles) in smallmouth bass [28]. Thus, the removal of intersex stimuli may lead to rapid recovery. 

This study was the first statewide assessment of surface water estrogenicity under base-flow conditions in the fall and one of the first to assess intersex prevalence and severity at multiple sites including impoundments within New Jersey. Information gained from this study is the first step in documenting exposure to EEDCs and understanding the potential broader impact this exposure could have on recreational fisheries. More information on the exposure, fate and effects of natural and synthetic estrogens in impoundments is needed as well as a broader overall assessment of fish health and reproductive viability in lakes, reservoirs and ponds. 

## Figures and Tables

**Figure 1 ijerph-17-02024-f001:**
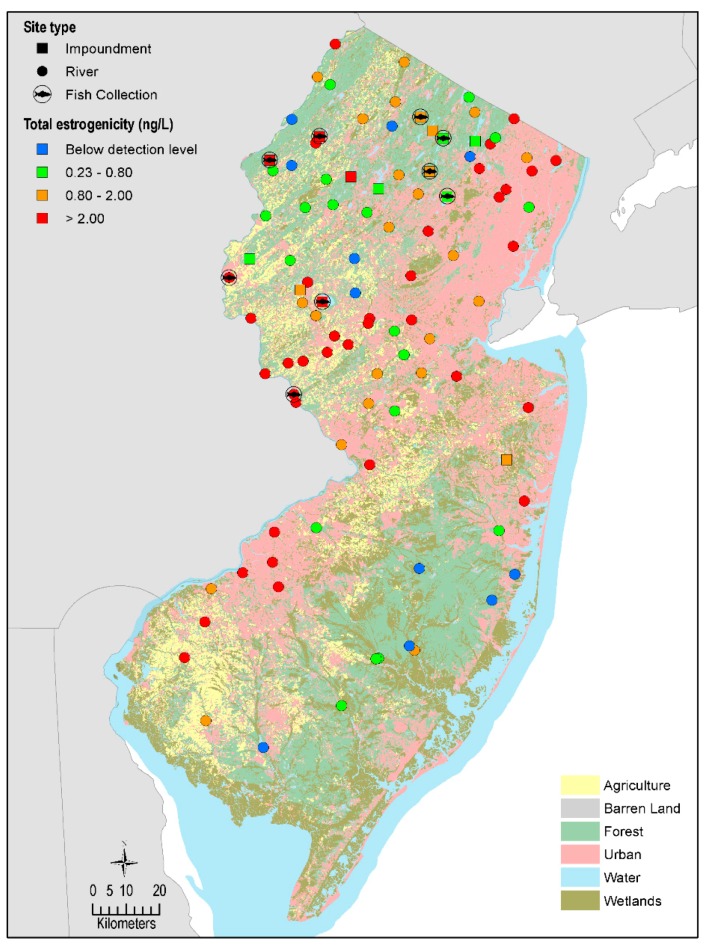
Surface water and fish sampling locations throughout New Jersey. Ranges of total estrogenicity (ng/L E2Eq_BLYES_) are binned as below detection (BD), low (0.23–0.80), medium (0.80–2.00) and high (>2.00). Values are those from the fall 2016 sampling. Dominant land-use designations are colormetrically depicted across the landscape.

**Figure 2 ijerph-17-02024-f002:**
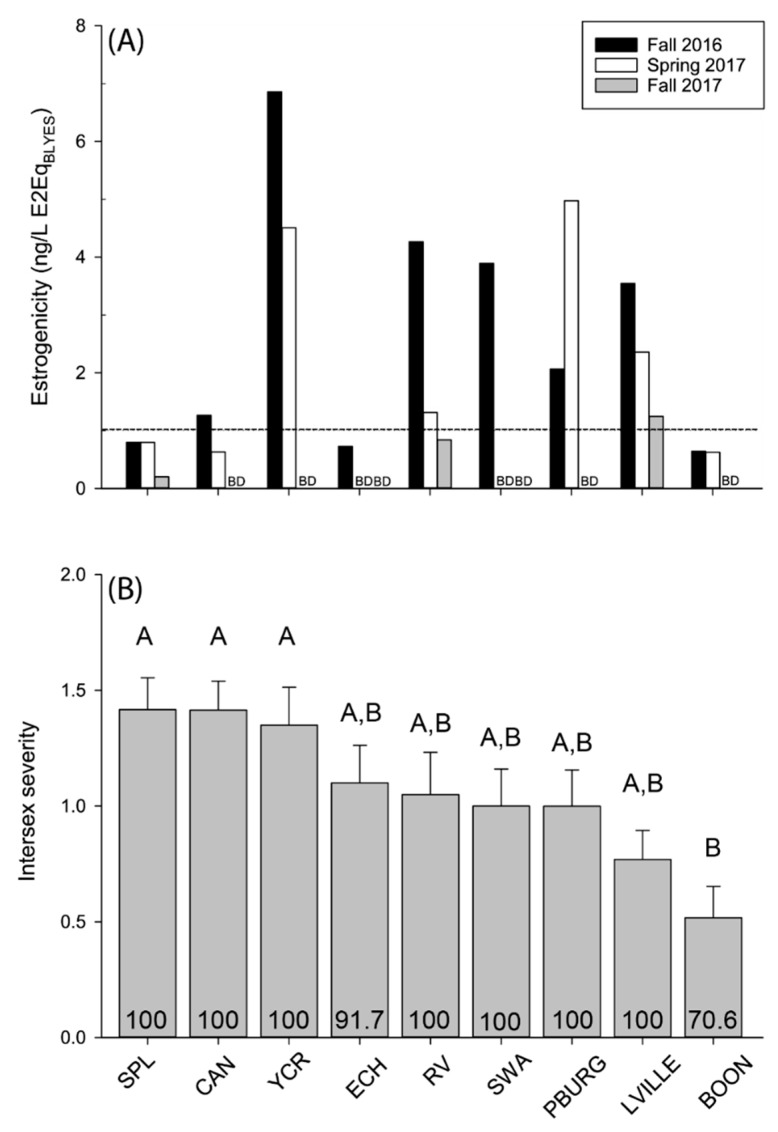
(**A**) Seasonal variation in EEQ_BLYES_ at the fish collection sites. (**B**) Intersex severity in male smallmouth bass collected from New Jersey during the spring of 2017. Intersex prevalence (%) is indicated at the base of each vertical bar. Error bars denote standard error of the mean. Statistical differences in intersex severity are indicated above vertical bars. Bars denoted with different letters indicate statistically significant differences (α = 0.05). Site abbreviations are defined in Table 1. Dotted line indicates the USEPA effects-based trigger value (EBT) of 1 ng/L E2Eq. BD indicates values below detection.

**Figure 3 ijerph-17-02024-f003:**
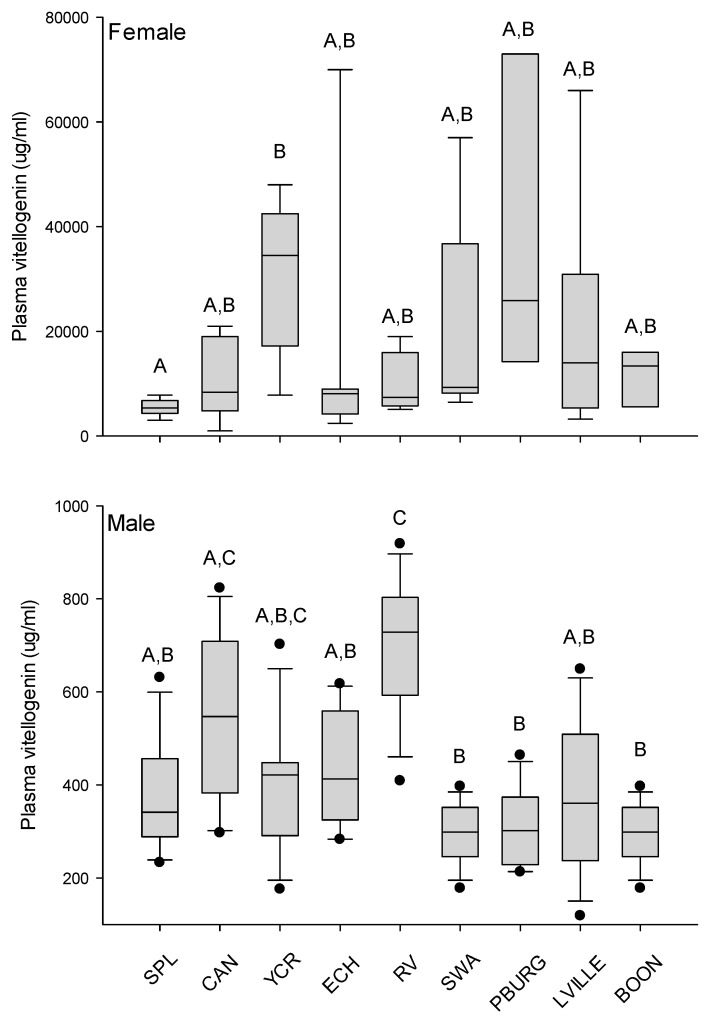
Plasma vitellogenin in female and male smallmouth bass collected in New Jersey during the spring of 2017. Sites are ordered along the x-axis based on decreasing intersex severity observed in male fish. Box and whisker plots with 95% confidence intervals. Black dots indicate outliers. Bars denoted with different letters indicate statistically significant differences (α = 0.05). Site abbreviations are defined in Table 1.

**Table 1 ijerph-17-02024-t001:** Sex-sorted morphometric data (averages) for smallmouth bass collected from New Jersey waters during 2017.

Site	Sex	Collection Dates	*n*	Age (years)	Length (mm)	Weight (g)	Condition Factor (Fulton’s K)
Round Valley Reservoir (RV)	M	4/5 and 4/12	12	5.6	323.8	394.5	1.15
	F	4/5/ and 4/12	8	5.1	312.6	330.1	1.08
Splitrock Reservoir (SPL)	M	4/12 and 4/19	12	3.8	332.3	484.8	1.27
	F	4/12 and 4/19	8	3.3	345.4	467.1	1.18
Echo Lake (ECH)	M	4/17	12	5.2	360.7	628.8	1.26
	F	4/17	8	5.0	339.0	566.1	1.29
Delaware River at Phillipsburg (PBURG)	M	5/3	11	3.7	278.9	321.1	1.27
	F	5/3	3	4.0	308.7	426.0	1.42
Swartswood Reservoir (SWA)	M	4/18	13	4.9	358.5	630.2	1.23
	F	4/18	7	5.6	360.9	695.1	1.30
Boonton Reservoir (BOON)	M	4/19	17	6.4	443.7	1327.4	1.45
	F	4/19	3	3.7	332.3	485.7	1.32
Yards Creek Reservoir (YCR)	M	4/20	12	8.9	419.3	968.3	1.31
	F	4/20	8	7.4	384.1	710.6	1.23
Delaware River at Lambertville (LVILLE)	M	4/28	13	4.5	352.4	644.0	1.35
	F	4/28	7	3.4	307.9	395.4	1.23
Canistear Reservoir (CAN)	M	4/17	13	4.3	308.0	453.0	1.16
	F	4/17	7	4.9	335.0	525.7	1.38

**Table 2 ijerph-17-02024-t002:** Spearman rank-order correlations among surface water estrogenicity (ng/L) and select landscape variables. Values in bold/italics are statistically significant (*p* ≤ 0.05).

Landscape Variable	All Sites (*N* = 101)	Rivers Only (*N* = 85)	Reservoirs Only (*N* = 16)
Percent Urban	***0.206***	***0.277***	−0.135
Percent Agriculture	0.0958	0.118	0.147
Percent Cultivated Crops	0.0761	0.0866	0.0678
Percent Forest	−0.188	−0.212	−0.0765
Percent Altered	***0.306***	***0.402***	−0.0176
Watershed Area	0.0557	0.128	−0.0294
Number of GW Discharge Permits	−0.0101	0.082	−0.385
Number of SW Discharge Permits	0.154	***0.249***	−0.130
Number of Contaminated Sites	0.157	***0.253***	0.059
Number of Landfills	0.0139	0.019	0.252

GW, groundwater; SW, surface water; altered, urban + agriculture.

**Table 3 ijerph-17-02024-t003:** Spearman rank-order correlations among fish health metrics including intersex severity, plasma vitellogenin and liver vitellogenin and select landscape variables. These data include river and reservoir sites (*N* = 113 male fish at nine sites). Values in bold/italics are statistically significant (*p* ≤ 0.05).

Landscape Variable	Plasma Vitellogenin	Liver Vitellogenin
Percent Urban	***−0.539***	***0.251***
Percent Agriculture	***−0.219***	0.0481
Percent Cultivated Crops	−0.0075	−0.0759
Percent Forest	***−0.229***	***−0.389***
Percent Altered	***−0.412***	***0.346***
Number of GW Discharge Permits	***−0.330***	−0.0386
Number of SW Discharge Permits	***−0.329***	−0.138
Number of Contaminated Sites	***−0.404***	−0.109

Altered, urban + agriculture.

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
