# Peer review of "Reconnaissance of Surface Water Estrogenicity and the Prevalence of Intersex in Smallmouth Bass (Micropterus Dolomieu) Inhabiting New Jersey"

_ijerph, 2020, doi:10.3390/ijerph17062024_

Round 1

Reviewer 1 Report

The manuscript entitled "Reconnaissance of surface water estrogenicity and the prevalence of intersex in smallmouth bass (Micropterus dolomieu) inhabiting New Jersey" is well designed and of interest to a wide range of scientists in the ecotoxicology field. The methodologies utilized in this paper appear appropriate and the dataset is robust. Some minor edits for clarity and flow are required (see below), as such I recommend that this manuscript be accepted with minor revisions.

line and comment

52 - generally includes the morphological presentation "of" testicular oocytes

52-53 - "This condition is termed intersex" - awkward phrasing, reword or combine with previous statement

55-57 - What are some examples of the types of land use and contaminants? Are there particular types of sites or contaminants that have not been clearly associated?

60 - 61 - "That research identified smallmouth..." - awkward phrasing, reword or combine with previous sentence

74 - "yeast based screening assay" - please provide some examples of previous studies that have utilized this assay to assess estrogenicity in surface waters 

98-99- Why were duplicates only collected for 19% of the sites? Please clarify this statement as it is unclear whether this was intentional and/or part of the study design

107-108- "Fish were collected from each of 9 locations within a maximally compressed time window as
108 logistically possible to circumvent natural biological differences based on seasonal change" - What was this time window?

110-112- "Fish collection, handling and euthanasia protocols were approved by the US Geological Survey, Leetown Science Center Institutional Animal Care and Use Committee" - please provide IACUC protocol identification number

113- Provide the specific MS-222 dose used for euthanasia

358-362- "Smallmouth bass are piscivorous and it seems plausible that ingestion of forage during high estrogen stages of the reproductive cycle may explain increases in baseline vitellogenin. Interestingly this is not a commonly acknowledged or investigated exposure route, even though ingestion of a single female prey item may lead to an exposure of many micrograms or estrogens" - I think a statement like this may lean too far towards conjecture, as intersex would be more widely observed if it was being caused by the consumption of ovulating female fish. Additionally, many steroid hormones expressed by females play an important role as pheromonal signals.

Author Response

52 - generally includes the morphological presentation "of" testicular oocytes

OK

52-53 - "This condition is termed intersex" - awkward phrasing, reword or combine with previous statement

OK, combined with previous sentence

55-57 - What are some examples of the types of land use and contaminants? Are there particular types of sites or contaminants that have not been clearly associated?

Examples are included in the discussion. The references in this specific sentence also point readers to the appropriate literature.

60 - 61 - "That research identified smallmouth..." - awkward phrasing, reword or combine with previous sentence

Revised sentence

74 - "yeast based screening assay" - please provide some examples of previous studies that have utilized this assay to assess estrogenicity in surface waters 

Included references for other studies that use this estrogenicity assay to assess estrogenic activity in surface waters

98-99- Why were duplicates only collected for 19% of the sites? Please clarify this statement as it is unclear whether this was intentional and/or part of the study design

Including duplicates for ~20% of samples is accepted convention for water chemistry-based sampling. This is essentially a technical replicate to QA/QC the process. It is not a true site replicate. 

107-108- "Fish were collected from each of 9 locations within a maximally compressed time window as logistically possible to circumvent natural biological differences based on seasonal change" - What was this time window?

The fish collections occurred between 4/5/17 and 5/3/17. This information is included in Table 1. We now refer to that table following our statement. 

110-112- "Fish collection, handling and euthanasia protocols were approved by the US Geological Survey, Leetown Science Center Institutional Animal Care and Use Committee" - please provide IACUC protocol identification number

The Leetown Science Center does not issue IACUC identification numbers at present. The IACUC protocol is associated with the study plan number. This information has been added to the acknowledgement section. Additionally, the IACUC protocol and approval documentation has been sent to the journal office.

113- Provide the specific MS-222 dose used for euthanasia

We have added this information

358-362- "Smallmouth bass are piscivorous and it seems plausible that ingestion of forage during high estrogen stages of the reproductive cycle may explain increases in baseline vitellogenin. Interestingly this is not a commonly acknowledged or investigated exposure route, even though ingestion of a single female prey item may lead to an exposure of many micrograms or estrogens" - I think a statement like this may lean too far towards conjecture, as intersex would be more widely observed if it was being caused by the consumption of ovulating female fish. Additionally, many steroid hormones expressed by females play an important role as pheromonal signals.

We disagree that this is conjecture. Intersex (defined as testicular oocytes) and elevated plasma vitellogenin are very different measures. There is plenty of evidence that oral ingestion of estrogens leads to increased plasma vitellogenin. We agree that the ingestion of prey items with high estrogen levels is unlikely to induce intersex. Our statement here is explicit to vitellogenin.   

Reviewer 2 Report

The manuscript entitled "Reconnaissance of surface water estrogenicity and the prevalence of intersex in smallmouth bass (Micropterus dolomieu) inhabiting New Jersey" makes a meaningful contribution to the field of ecotoxicology. The introduction is well-written, summarizes recent research related to the topic and establishes the originality of the research aims by demonstrating the need for investigations in the topic area. The utilized methodologies are appropriate. The results appear solid and the studied dataset was large enough to draw conclusions from the results.

Minor revisions

106 – smallmouth bass were sampled from 9 locations - please consider adding (Figure 1)

118 – ... Mini (Benchmark Scientific) - please consider adding City, Country

138 – (Applied Biosystems) - please consider adding City, Country

Author Response

The manuscript entitled "Reconnaissance of surface water estrogenicity and the prevalence of intersex in smallmouth bass (Micropterus dolomieu) inhabiting New Jersey" makes a meaningful contribution to the field of ecotoxicology. The introduction is well-written, summarizes recent research related to the topic and establishes the originality of the research aims by demonstrating the need for investigations in the topic area. The utilized methodologies are appropriate. The results appear solid and the studied dataset was large enough to draw conclusions from the results.

Minor revisions

106 – smallmouth bass were sampled from 9 locations - please consider adding (Figure 1)

Thank you. This is a good idea. We now reference this figure in this statement. 

118 – ... Mini (Benchmark Scientific) - please consider adding City, Country

OK

138 – (Applied Biosystems) - please consider adding City, Country

OK